# Monitoring the Influence of Industrialization and Urbanization on Spatiotemporal Variations of AQI and PM$_{2.5}$ in Three Provinces, China

Hu Chen , Guoqu Deng * and Yiwen Liu

School of Management, Henan University of Science and Technology, Luoyang 471023, China
* Correspondence: dengguoqu@haust.edu.cn

**Abstract:** With the rapid development of industrialization and urbanization, atmospheric pollution research is vital for regional sustainable development and related policies formulated by the government. Previous studies have mainly studied a single evaluation method to analyze the air quality index (AQI) or single air pollutant. This research integrated the Spearman coefficient (SC) correlation analysis, a random search (RS) algorithm and an excellent extreme gradient boosting (XGBoost) algorithm to evaluate the air pollution influence of industrialization and urbanization (APIIU). Industrialization, urbanization and meteorological indicators were used to measure the influence degree of APIIU on AQI and particulate matter 2.5 (PM$_{2.5}$), respectively. The main findings were: (1) the APIIU-AQI and APIIU-PM$_{2.5}$ of Henan Province, Hubei Province and Hunan Province had significant changes from 2017 to 2019; (2) the value of square of determination coefficient of real value ($R^2$), the root mean square error (RMSE) and the mean absolute percentage error (MAPE) of APIIU-AQI and APIIU-PM$_{2.5}$ in three provinces predicted by the SC-RS-XGBoost were 0.945, 0.103, 4.25% and 0.897, 0.205, 4.84%, respectively; (3) the predicted results were more accurate than using a SC-XGBoost, RS-XGBoost, traditional XGBoost, support vector regression (SVR) and extreme learning machine (ELM).

**Keywords:** air pollution; industrialization and urbanization; SC-RS-XGBoost; air quality; particulate matter

## 1. Introduction

Air quality has become a more significant problem with the development of industrialization and urbanization [1–4]. The remarkable economic growth has resulted in serious environmental issues [5,6]. The emissions of air pollutants from industrial and motor vehicles are currently the most important environmental risk to human health [7]. According to an assessment report from the World Health Organization, urban air pollution has resulted in more than two million deaths every year [8,9]. In the past 20 years, the European Union has made great policy progress in atmospheric emissions and air quality. However, air pollution continues to have a serious influence on the health of people who live in Europe's urban areas [10,11]. As one of the rapidly developing countries, China has experienced rapid economy growth in the past several decades. As a result, industrial activities, urban expansion and human engineering activities have produced a high intensity of air pollutant emissions in China [12–16]. Hence, reducing air pollution is a critical measure for environmental protection and sustainable development [17].

From 2010, China has actively implemented the Clean Air Action to deal with air pollution, and many pollutants' emissions have decreased since then [18]. To effectively assess air quality and provide guidance for outdoor activities, the Chinese Ministry of Environmental Protection (MEP) has adopted and developed an air quality index (AQI) system in 2012 [19–21]. The existing related research works elaborately documented that the six major air pollutants were particulate matter 2.5 (PM$_{2.5}$), particulate matter 10 (PM$_{10}$), sulfur oxides (SO$_2$), nitrogen oxides (NO$_2$), carbon mono oxide (CO) and ozone (O$_3$). The

AQI converted six pollutants into a comparable and easily understandable index [22–24]. Despite improvements over several years, China, with a sizeable economic aggregate and dense population, continues to cause air pollution. Additionally, the economic development model with high energy consumption and low efficiency is an important factor that leads to air pollution in China [25–28]. The MEP released the 2019 China Ecological and Environmental Bulletin in June 2020, which reported that only 157 cities reached the air quality standards among the 337 cities. Accordingly, a total of 1666 days of heavy pollution and 452 days of severe pollution were reported in 337 cities during 2019. Air pollution has caused an obviously serious impact on residents' daily lives.

Considering the above existing problems, the prevention and prediction of air pollution have become the focus of researchers all over the world. Nowadays, the related research works have mainly focused on the precise prediction of AQI and other air pollutant levels, which was important for the early air quality warning and policymakers' work. Many AQI forecasting models have been presented in recent years, including physical models, statistical models and hybrid models. Among them, the physical methods were more complicated and consuming too much time, and statistical models have been proven to outperform the physical methods [29–33]. The hybrid models combined data decomposition, feature extraction and optimization methods to improve the forecasting performance. Munawar et al. [34] adopted a neuro-fuzzy inference system to forecast AQI and presented a case study of Lahore city of Pakistan. Wang et al. [35] used a hybrid forecasting model combining a two-phase decomposition technique and extreme learning machine (ELM) optimized by differential evolution (DE) algorithm to forecast AQI. Li et al. [36] used hybrid models based on the auto regressive integrated moving average model (ARIMA), optimized extreme learning machine (OELM) and fuzzy time series model (FTSM) to forecast the reconstructed series in the dynamic integration forecasting module. By employing time varying parameters to dynamically combine the forecasting results, the accuracy was superior to other forecasting models. Phruksahiran [37] adopted an ensemble prediction methodology and used the additional predictor variables for predicting AQI at the hourly level, which combined the geographically weighted predictor method (GWP) and the related machine-learning algorithms. The results showed that the hybrid model enhanced the accuracy of AQI prediction. Despite the better accuracy of the above models compared to the traditional prediction methods, they lacked the prediction of concentration of a single pollutant [38–42].

With the rapid development of artificial intelligence, more accurate models have been developed. Due to the advantages of robustness and self-adaptability, these models have been regarded as effective and outstanding approaches for forecasting air pollutant concentrations. Feng et al. [43] analyzed and accurately forecasted the atmospheric pollutants in Hangzhou using a hybrid model based on recurrent neural network (RNN) and random forest (RF). Franceschi et al. [44] adopted a hybrid model based on artificial neural networks (ANN), principal component analysis (PCA) and K-means clustering to forecast the concentrations of $PM_{2.5}$ and $PM_{10}$ in Bogotá, Colombia. The above model had the ability to accurately predict air pollutants, which could be used for early warnings of high air pollution. Gao et al. [45] investigated the feasibility of using a hybrid model, which was based on ANN and Monte Carlo simulations (MCS) with meteorological parameters to predict the concentration of $O_3$ in the urban area of Jinan, China. Li et al. [46] designed a novel long short-term memory neural network (LSTM) extended model to predict air pollutant concentration. It considered historical air pollutant data, meteorological data and time stamp data, which were merged into the model to enhance the accuracy. Liu et al. [47] proposed a hybrid neural network model to forecast the concentration of $PM_{2.5}$. To obtain a much better performance and forecasting outputs, ELM with weighted regularization, whose parameters were optimally determined by the ANN algorithm, was used. Pak et al. [48] used a hybrid model based on convolutional neural networks (CNN) and LSTM (CNN-LSTM) to efficiently extract the inherent features of huge air quality and meteorological data; the model was proposed and used for predicting the concentration of $O_3$ in Beijing

City and exhibited a better stability and prediction performance. Zhou et al. [49] adopted the Gaussian process mixture (GPM) model based on an iterative learning algorithm to predict the air pollutants' concentrations. Yu et al. [50] developed a dynamic model based on ELM to forecast the concentrations of $SO_2$ and $NO_2$. Using the quantum genetic algorithm (QGA) to optimize the connection weight threshold of the ELM, it contributed to increasing the prediction performance. Middya et al. [51] found that several models, such as the bi-directional LSTM (Bi-LSTM) and convolutional LSTM (Conv-LSTM), achieved a high forecasting accuracy for the majority of air pollutants. Sun et al. [52] established a new model based on the stacking-driven ensemble (SDE) and two kinds of input selection methods to forecast the hourly concentration of $PM_{2.5}$ and found that the proposed model had a higher predicting accuracy, much better performance and more robust forecasting ability. Ribeiro [53] used Bayesian, econometrics and machine-learning models applied to predict the future concentration of $SO_2$ emissions, respectively. Finally, the machine-learning models had better generalization power than traditional methods. Although previous research works had superiority in predicting the concentrations of air pollutants, there were few studies of air quality based on analyzing the influence of relevant indicators of air quality in the process of industrialization and urbanization. Rapid economic growth and industrialization have decreased air quality in developing countries. The urbanization scale and high population density have increased air pollutant emissions through various human daily travel and life activities [54–58].

The above literature demonstrates the research on AQI and air pollutant concentrations with different methods. Currently, there is no universally agreed method of constructing an indicators system for the assessment of air pollution influence of industrialization and urbanization (APIIU). The definition of an APIIU indicators system is variable, and it can hold multiple dimensions and present different specificities depending on the province, time span or assessment target variable for air pollution. This uncertainty makes it difficult to identify consistent research methods for analysis and prediction. In this paper, considering the availability of data and the cumulative effect of development of industrialization and urbanization, our approach was to choose annual data. Since the study area and time span were selected from 49 cities in three provinces from 2017 to 2019, the above data belonged to a small sample. Extreme gradient boosting (XGBoost), support vector regression (SVR) and ELM have been proven to possess good prediction performance for small samples in past research [59–65]. Additionally, since the construction of the indicators system was mainly based on previous research, the Spearman coefficient (SC) was used for analyzing the constituent indicators of APIIU of AQI and $PM_{2.5}$ to select indicators with a large correlation coefficient. When using XGBoost and SVR to predict data, internal parameters affect the prediction accuracy. Hence, the parameters need to be optimized to improve the performance results. XGBoost includes the general, booster and learning task parameters. Many types and quantities of parameters needed to be optimized in this research, and we adopted a random search (RS) algorithm to optimize the parameters of XGBoost and SVR [66–68]. On the one hand, the algorithm was selected to consider the efficiency of optimizing XGBoost. On the other hand, it was used for comparing the prediction performance of RS-XGBoost and RS-SVR.

This study firstly considered the industrialization, urbanization, population, regional gross domestic product (GDP) and meteorological indicators as the APIIU indicators. Then, we constructed an APIIU indicators system from the aspects of economy, society and natural environment. This research was divided into four sections. The first section introduces the research review. The Materials and Methods section introduces the research region, data sources, indicators system and the principle of the SC-RS-XGBoost. The Analytical Results section introduces the determination of each indicator's weight and calculation of APIIU, and the final section offers a discussion, our conclusions and the limitations of this research.

## 2. Materials and Methods

### 2.1. Data

2.1.1. Research Region

The research domain covers 49 cities composed of 18 cities in Henan Province, 17 cities in Hubei Province and 14 cities in Hunan Province, which is shown in Figure 1. Henan Province is located in the central part of eastern China and the middle and lower reaches of the Yellow River. It is bounded by latitude 31°23′–36°22′ N and longitude 110°21′–116°39′ E, with a total area of 167,000 square kilometers. Hubei Province is located in the central part of China with a total area of 185,900 square kilometers, between latitude 29°0′53″–33°03′ N and longitude 108°21′42″–116°07′50″ E, in the middle reaches of the Yangtze River. Hunan Province is located in the central part of China with a total area of 211,800 square kilometers, between latitude 24°38′–30°08′ N and longitude 108°47′–114°15′ E. At the end of 2021, the resident population of Henan Province was 98.8 million, and the regional GDP was over CNY 58,887 billion. The resident population of Hubei Province was 58.3 million, and the regional GDP was over CNY 50,012 billion. The resident population of Hunan Province was 66.2 million, and the regional GDP was over CNY 46,063 billion. In these three provinces, the proportion of secondary industry is 41.3%, 37.9% and 39.4%, respectively. The related research works showed that the economic scale and high proportion of secondary industry had a significant impact on air quality and environmental pollution. In the 2021 census data, the population of Henan Province, Hubei Province and Hunan Province, respectively, ranked third, seventh and tenth in thirty-four provinces in China. The construction of urbanization has resulted in a decrease in urban green vegetation coverage, which directly contributed to air pollution. The higher proportion of secondary industry and the larger population have resulted in more air pollutant emissions and AQI exceeding the standard. Hence, it is necessary to study air quality in the three provinces. The geographic locations of the research regions are shown in Figure 1.

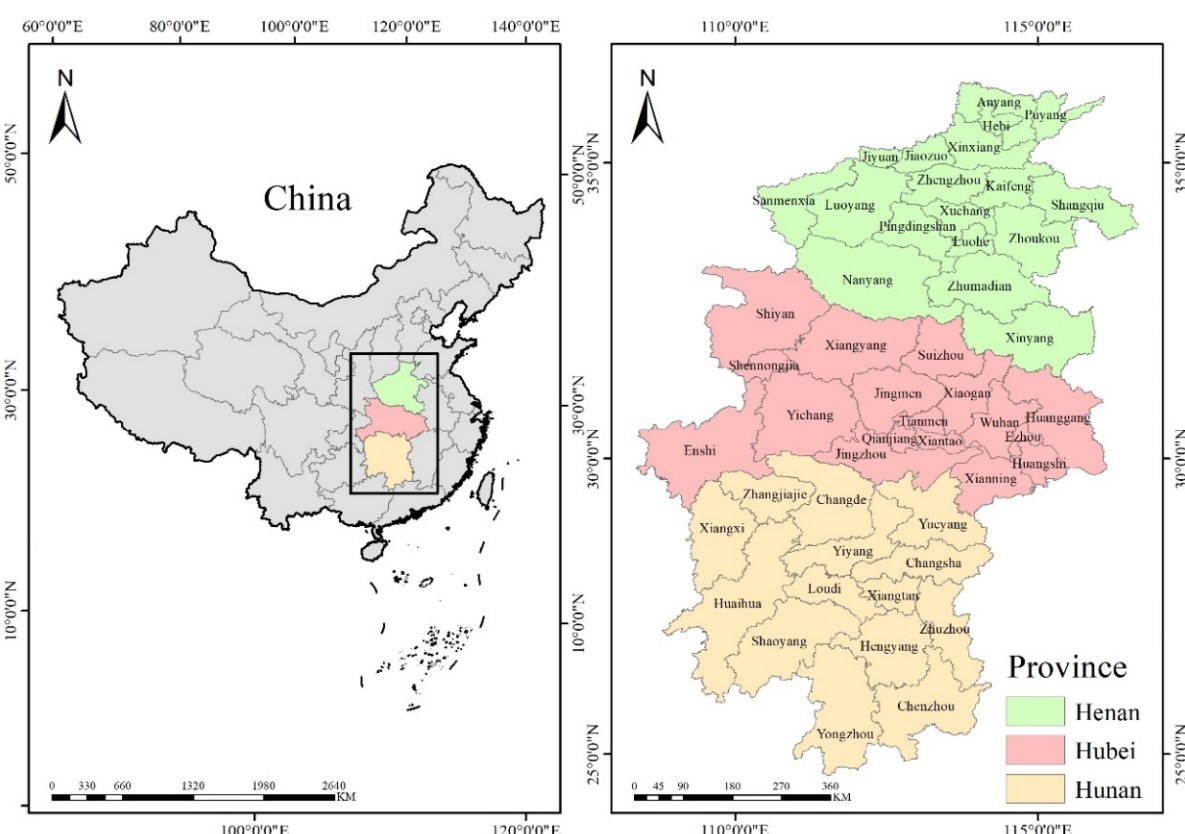

**Figure 1.** Regional geographic locations distribution map of APIIU research.

2.1.2. Data Sources

The data on the regional development of industrialization and urbanization used in this study include regional GDP, the GDP of secondary industry, coal consumption, exhaust emissions, population of city jurisdiction, total city population, city jurisdiction areas, administrative land area, density of population and so on. Due to yearbook data generally publishing the annual data of the previous year, the above data range was from 31 December 2018 to 31 December 2020 and was sourced from the National Statistics Administration and the China City Statistical Yearbook (https://data.stats.gov.cn/index.html/ accessed on 2 May 2022). The meteorological data included annual relative humidity, average temperature, rainfall data and time length of sunshine (http://data.cma.cn/dataService/cdcindex/datacodel/ accessed on 3 May 2022). The average annual AQI in 18 cities from 2017 to 2019 were calculated from the daily data from the air quality online monitoring platform (https://www.aqistudy.cn/historydata/ accessed on 5 May 2022). The annual average PM$_{2.5}$ were calculated from the daily data, which were from the US Embassy's Air Quality Report (https://www.airnow.gov/ accessed on 5 May 2022). In this paper, by using PyCharm 2021 in Python, Anaconda 3, SPSS Statistics 26.0 and ArcGIS 10.2, the average annual AQI changing trend is shown in Figure 2a–c. In addition, the relevant statistical analysis, including maximum value and standard deviation of the annual concentrations of PM$_{2.5}$ in every city in three provinces from 2017 to 2019, is shown in Tables S1–S3.

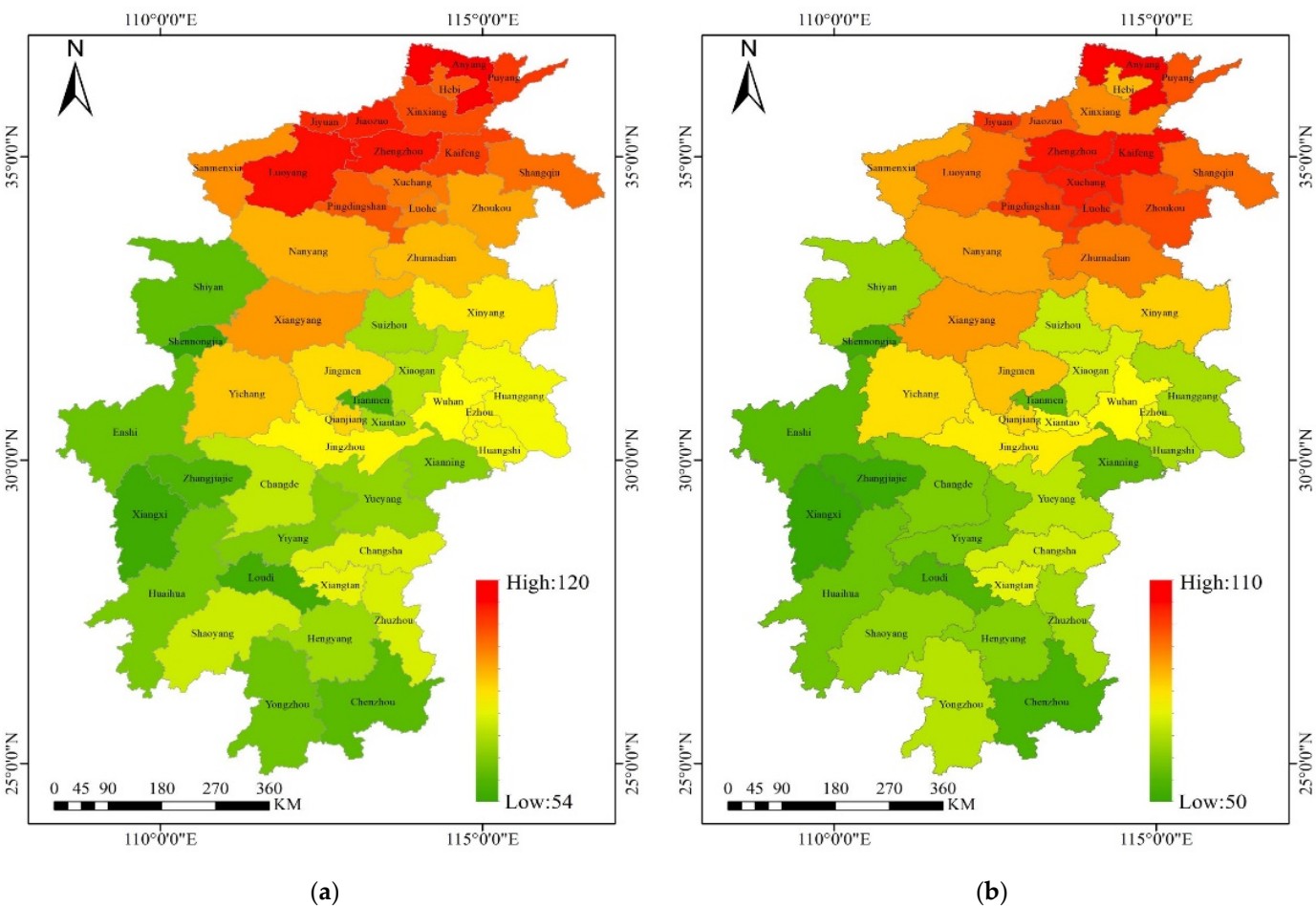

(**a**)　　　　　　　　　　　　　　　　(**b**)

**Figure 2.** *Cont.*

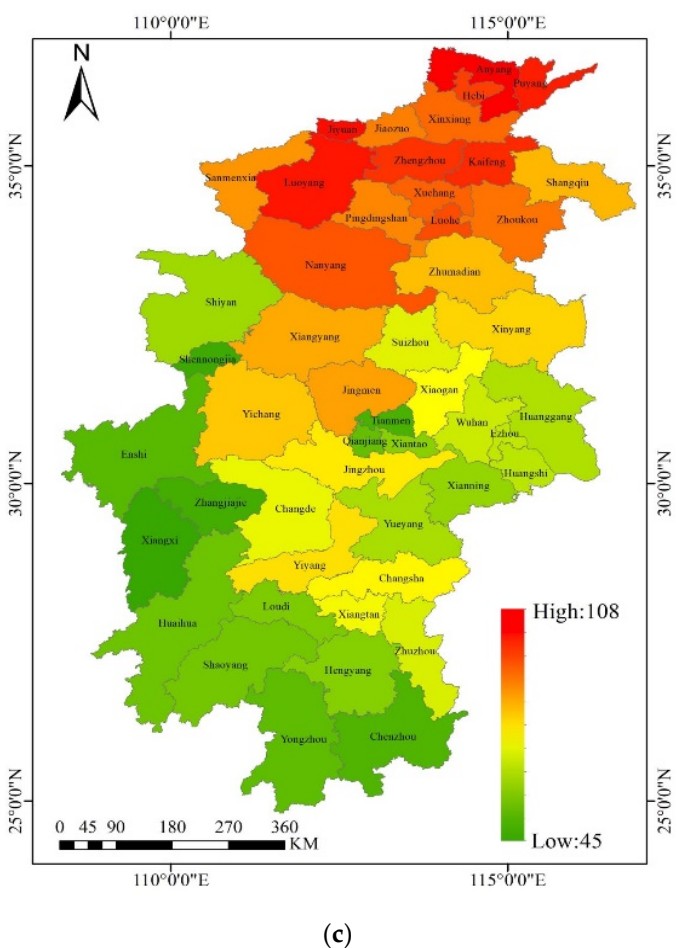

(**c**)

**Figure 2.** Annual average AQI changing trend in every province from 2017 to 2019: (**a**) Henan Province, Hubei Province and Hunan Province in 2017; (**b**) Henan Province, Hubei Province and Hunan Province in 2018; (**c**) Henan Province, Hubei Province and Hunan Province in 2019.

On the whole, the annual average AQI of Henan Province, Hubei Province and Hunan Province have slowly decreased year by year from 2017 to 2019 in some cities. However, the AQI of some cities in Henan Province have not decreased significantly. For instance, the annual average AQI in Zhengzhou, Anyang and Xinxiang in Henan Province have consistently exceeded 100 in the past three years. The annual average AQI in Xiangyang and inmen in Hubei Province have consistently exceeded 90 in the past three years. The annual average AQI in Changsha, Yueyang, Xiangtan and Zhuzhou in Hunan Province have consistently exceeded 70 in the past three years. Additionally, the annual average AQI in Yiyang in Hunan Province increased year by year from 2017 to 2019. To clearly observe the changing trend of $PM_{2.5}$ in three provinces from 2017 to 2019, the annual average $PM_{2.5}$ data of every city were collected and calculated. The annual average $PM_{2.5}$ changing trend is shown in Figure 3a–c.

The annual average concentrations of $PM_{2.5}$ of Henan Province, Hubei Province and Hunan Province have decreased year by year from 2017 to 2019. For Henan Province, the past three years have witnessed the annual average concentrations of $PM_{2.5}$ decrease by more than 15% in Luoyang, Sanmenxia and Xinyang. However, other cities have consistently exceeded 40 $\mu g/m^3$ in 2019. For Hubei Province, Huangshi, Enshi and Shennongjia, with relatively pleasant natural environment and low level of urbanization, had a concentration less than 30 $\mu g/m^3$ in 2019, which was the same as the change trend of AQI in recent years. For Hunan Province, Zhangjiajie, Chenzhou and Xiangxi had a concentration less than 30 $\mu g/m^3$ in 2019. Among the three cities, the annual average concentration of Zhangjiajie had dropped by more than 18% compared with 2017. Although China has

issued related policies to protect the environment and manage air pollutant emissions in recent years, with the development of industrialization and urbanization and the increase in the population scale, the annual average concentrations of $PM_{2.5}$ of most cities are still more than 30 μg/m$^3$, and even some cities have consistently exceeded 60 μg/m$^3$ in 2019. Hence, it is very necessary to carry out the relevant environmental research of AQI and particulate matter, providing the government with a feasibility analysis report.

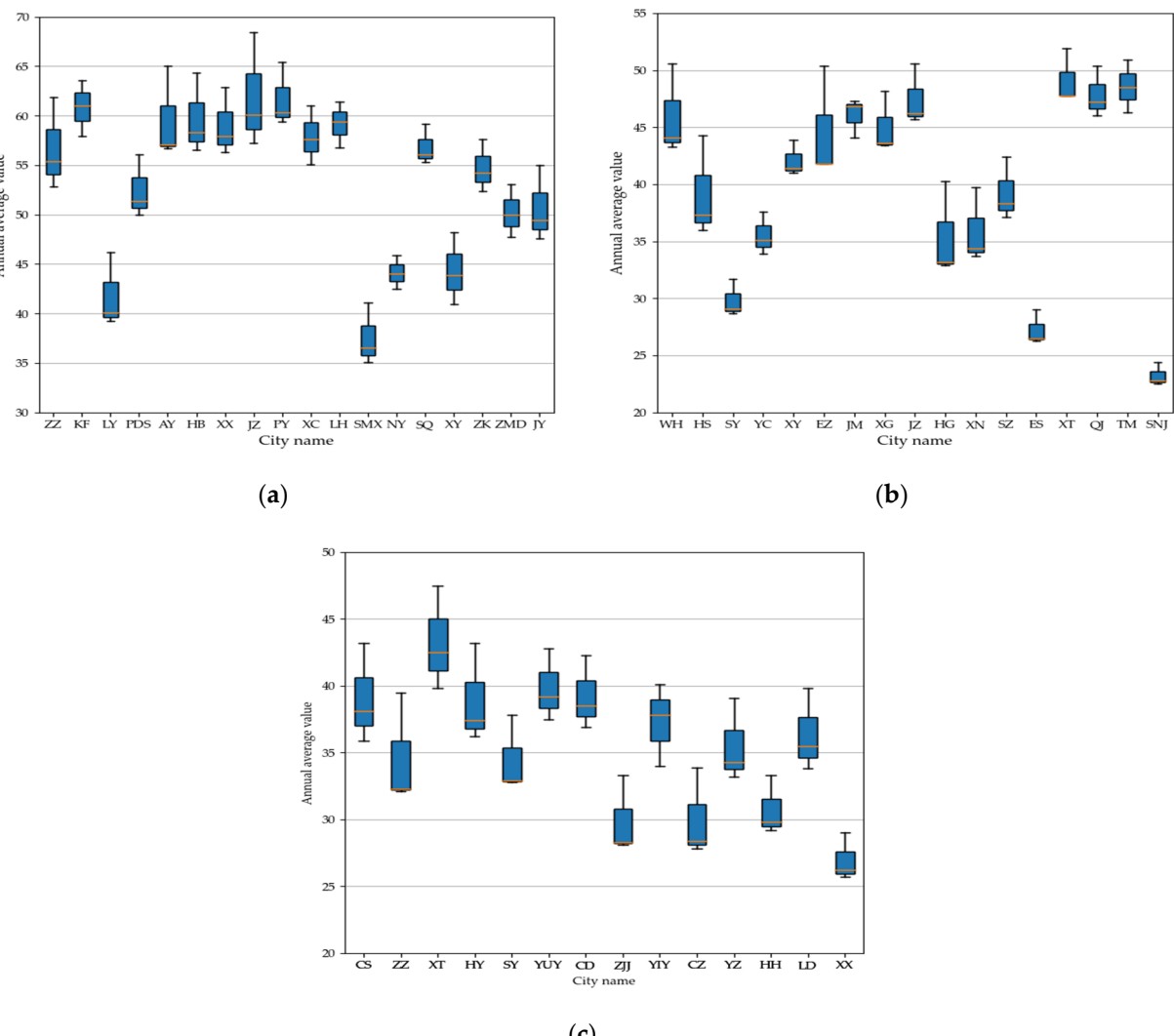

**Figure 3.** The changing trend of APIIU-AQI from 2017 to 2019: (**a**) Henan Province: Zhengzhou—ZZ, Kaifeng—KF, Luoyang—LY, Pingdingshan—PDS, Anyang—AY, Hebi—HB, Xinxiang—XX, Jiaozuo—JZ, Puyang—PY, Xuchang—XC, Luohe—LH, Sanmenxia—SMX, Nanyang—NY, Shangqiu—SQ, Xinyang—XY, Zhoukou—ZK, Zhumadian—ZMD, Jiyuan—JY; (**b**) Hubei Province: Wuhan—WH, Huangshi—HS, Shiyan—SY, Yichang—YC, Xiangyang—XY, Ezhou—EZ, Jingmen—JM, Xiaogan—XG, Jingzhou—JZ, Huanggan—HG, Xianning—XN, Suizhou—SZ, Enshi—ES, Xiantao—XT, Qianjiang—QJ, Tianmen—TM, Shennongjia— SNJ; (**c**) Hunan Province: Changsha—CS, Zhuzhou—ZZ, Xiangtan—XT, Hengyang—HY, Shaoyang—SY, Yueyang—YUY, Changde—CD, Zhangjiajie—ZJJ, Yiyang—YIY, Chenzhou—CZ, Yongzhou—YZ, Huaihua—HH, Loudi—LD, Xiangxi—XX.

### 2.1.3. Indicators System

The regional assessment indicators system of APIIU constructed from the perspective of four aspects is shown in Table 1.

**Table 1.** Primary and secondary indicators.

| Primary Indicators | Secondary Indicators | Reference Source |
|---|---|---|
| Industrialization indicators | $X_1$: Regional GDP (CNY 100 million) | [1,2,4,54,57] |
| | $X_2$: Regional GDP of secondary industry (CNY 100 million) | |
| | $X_3$: Proportion of secondary industry (%) | [1,2,4,54,57] |
| | $X_4$: Square of proportion of secondary industry (%) | |
| | $X_5$: Coal consumption (100 million ton) | |
| | $X_6$: Coal consumption per land area (100 million ton/km$^2$) | |
| | $X_7$: Square of coal consumption per land area (100 million ton/km$^2$) | |
| | $X_8$: Exhaust emissions (ton) | |
| | $X_9$: Density of exhaust emissions (ton/km$^2$) | |
| | $X_{10}$: Square of density of secondary industry (ton/km$^2$) | |
| Urbanization indicators | $X_{11}$: Population of city jurisdiction (10 million people) | [1,12,33,58,68] |
| | $X_{12}$: Total city population (10 million people) $X_{13}$: Proportion of population (%) $X_{14}$: Square of proportion of population (%) | |
| | $X_{15}$: City jurisdiction areas (km$^2$) | |
| | $X_{16}$: Density of population (10 million people/km$^2$) | |
| | $X_{17}$: Square of density of population (10 million people/km$^2$) | |
| | $X_{18}$: Per capita GDP (CNY 10,000/people) $X_{19}$: Administrative land areas (km$^2$) | |
| Meteorological indicators | $X_{20}$: Annual average relative humidity (%) | [45,46,48,58–66] |
| | $X_{21}$: Annual average temperature (°C) | |
| Meteorological indicators | $X_{22}$: Annual average rainfall (mm) $X_{23}$: Time length of sunshine (hour) | [45,46,48,58–66] |
| | $X_{24}$: Wind speed (m/s: Annual average wind speed at 70 m–80 m altitude) | |
| Type of APIIU | $X_{25}$: APIIU of AQI $X_{26}$: APIIU of PM$_{2.5}$ | [22–24,35–37,52] |

　　　　Based on the existing related research results, considering the current situation regarding air pollution and the availability of data, this study constructed an indicators system of APIIU assessment using the industrialization, urbanization and meteorological indicators. The industrialization indicators were directly responsible for air pollutant emissions. Previously published studies have shown that the major air pollutant emissions were from the daily industrial operation of the machines. Therefore, the regional GDP, GDP of the secondary industry, coal consumption and exhaust emissions were used as direct or indirect influencing indicators of APIIU. Urbanization indicators included the city's development level, scale and population. The occurrence of air pollution was closely related to the city's development level and scale. With population growth and urban development, more and more building construction areas contributed to the decrease in the city's green vegetation areas, which increased the concentrations of dust and particulate matter in the air and reduced the adsorption capacity of green plants for air pollutants. The higher density of the

regional population living in urban areas caused more environmental pollution problems and air pollutant emissions. The annual average air humidity and the annual average rainfall could decrease air pollution to some extent. Higher emissions of industrial and residential pollutants caused higher annual average temperature. $PM_{2.5}$ refers to particulate matter with aerodynamic equivalent diameter of less than or equal to 2.5 microns. Owing to long residence time and conveying distance, it has a greater impact on human health and the quality of the atmospheric environment. $PM_{10}$ is inhalable particulate matter with aerodynamic equivalent diameter of less than or equal to 10 microns, which refers to the general term for solid and liquid particles floating in the air. $PM_{10}$ can enter the upper respiratory tract directly and endanger human lung health. $PM_{10}$ in many cities were positively correlated with $PM_{2.5}$, but $O_3$ was negatively correlated with $PM_{2.5}$. Related studies conducted in other regions also showed that the concentrations of $SO_2$, $NO_2$ and CO have increased year by year with the formulation of relevant policies. For example, the most obvious feature was that the occurrence of acid rain has reduced in the past many years [69,70]. Hence, this research selected the AQI and $PM_{2.5}$ as the objectives of APIIU; industrialization indicators, urbanization indicators and meteorological indicators were selected to evaluate the influence degree of APIIU on AQI and $PM_{2.5}$.

### 2.2. The Hybrid SC-RS-XGBoost Model

#### 2.2.1. The Principle of SC

In statistics, the SC is a nonparametric measure of the dependence of two variables. It uses a monotonic equation to evaluate the correlation of two statistical variables. If there are no repeated values in experimental data, the value of SC is 1, or $-1$ when the two variables are in a perfect monotonical correlation. SC is also referred to as level correlation, which is the observed data being replaced by the level. The SC indicates the direction of correlation between independent variable $x$ and dependent variable $y$. If $y$ tends to increase with $x$ increases, the SC is positive. If $y$ tends to decrease with $x$ increases, the SC is negative. When SC is zero, it indicates that $y$ exhibits no trend with $x$ increases. The absolute value of SC increases when $x$ and $y$ become closer and closer [66–68]. The SC can reflect the direction and degree of the change trend between the 2 random variables, which is calculated as the difference of equal magnitude. The most notable feature is that it need not consider the sample size or overall distribution characteristics of the variables; the SC of two random variables can be expressed as Equation (1):

$$\rho(x,y) = \frac{\sum_{i=1}^{n}(x_i - \overline{x})(y_i - \overline{y})}{\sqrt{\sum_{i=1}^{n}(x_i - \overline{x})^2(y_i - \overline{y})^2}} \tag{1}$$

where $\rho$ is the SC between $x$ and $y$, $n$ is datasets sample size, $\overline{x}$ and $\overline{y}$ is the average value of $x$ and $y$.

In fact, the connection between the variables is irrelevant. The principle is to sort the data of two variables and calculate the linear correlation analysis using the value of rank difference obtained after sorting, as shown in Equation (2):

$$\rho = \frac{6\sum_{i=1}^{n} d^2}{n(n^2 - 1)} \tag{2}$$

where $n$ is datasets sample size, $d$ is the rank difference value sorted by 2 variables.

#### 2.2.2. The Principle of RS

The grid search (GS) is an exhaustive optimization algorithm, and the searched parameters are defined in the space of the uniform grids. All the nodes in the grid are then evaluated to identify the global minimum. Finally, the grid search finds the global minimum of all the nodes in the parameter grid. By approaching the optimal point in the next step, a finer grid is defined and gradually approaches the optimal point in the searching pa-

rameter space. However, the optimization process is more time consuming and inefficient. The RS algorithm selects the specific number of 1 random value per 1 hyperparameter to reduce the computation of the hyperparameter search, shorten the optimization time and improve the model performance. It is performed in a random manner in the spatial distribution, and the RS algorithm samples it as 1 distribution [59–61]. In this research, RS was applied to optimize XGBoost.

### 2.2.3. The Principle of XGBoost

The integrated learning method refers to combining multiple learning models to obtain better results and a stronger generalization ability. XGBoost is a scalable tree boosting system that is an ensemble method that aims to aggregate weak learning models to form a stronger and more robust estimator in an iterative fashion. The residual of the previous estimator will be used to learn and optimize the loss function during each iteration. A binary decision tree called a classification and regression tree (CART) is selected as a basic learner, and regularization is added to the loss function for improvement and to avoid overfitting in XGBoost [62–65]. XGBoost is a highly flexible and versatile tool that can solve most regression problems and objective functions created by users. The SC-RS-XGBoost algorithm used in this paper is shown in Algorithm 1.

---

**Algorithm 1:** SC-RS-XGBoost

---

**Input:** $D_{m \times n} = \{(X_i)\}$ ($X_i \in R^m$, $i = 1, 2, \ldots , n$), original data with $n$ samples and $m$ feature variables
**Output:** $D_{l \times n} = \{(X_i)\}$ ($X_i \in R^l$, $i = 1, 2, \ldots , n$), using SC to screen correlation coefficients greater than 0.3 based on $D_{m \times n}$ ($l < n$)
**Input:** Objective function of RS, $f(X,Y) = g(X) \, h(Y)$
  Set default values and value ranges of parameters to be optimized
Set the threshold of mean squared error (MSE)
**Output:** Every parameter value when $f(X,Y)$ reaches the maximum value

---

**Input:** $D_{l \times n} = \{(X_i)\}$ ($X_i \in R^l$, $i = 1, 2, \ldots , n$)
$I$, instance set of current node
$d$, feature dimension
$Gain \leftarrow 0$
$G \leftarrow \sum_{i \in I} g_i$, $H \leftarrow \sum_{i \in I} h_i$
**for** k = 1 *to* T **do**
    $G_L \leftarrow 0, H_L \leftarrow 0$
**for** $j$ *in sorted*($I$, *by* $x_{jk}$) **do**
        $G_L \leftarrow G_L \, g_j, H_L \leftarrow H_L \, h_j$
$G_R \leftarrow G - G_L, H_R \leftarrow H - H_L$
        score $\leftarrow max \, (score, \frac{G_L^2}{H_L \, \lambda} \, \frac{G_R^2}{H_R \, \lambda} - \frac{G^2}{H \, \lambda})$
    **end**
**end**
**Output:** Split with max score

---

The XGBoost algorithm includes three types of parameters: general, booster and learning task. There are relatively many types of parameters, and the selection of parameter combinations directly affects the accuracy of boosting tree regression prediction. In machine learning, a reasonable parameter selection can improve the prediction accuracy of the model to a large extent. In this study, the RS optimization algorithm was used to optimize some parameters of XGBoost, and the other parameters selected appropriate default values according to the training process. The range of parameter values and final parameter determination optimized by the RS algorithm are shown in Table 2.

**Table 2.** Optimized parameters and value range of XGBoost.

| Parameter Name | Parameter Type | Parameter Definition | Parameter Default Value | Value Range |
|---|---|---|---|---|
| $P_1$: max_depth | Booster | Maximum depth of tree | 5 | [1,20] |
| $P_2$: learning_rate | Booster | Learning rate | 0.2 | [0,1] |
| $P_3$: reg_gamma | Booster | Adjusting the penalty term, specifying the minimum loss function decreases when the node is divided | 0.01 | [0,0.5] |
| $P_4$: reg_alpha | Booster | Regularization coefficient used to adjust *L1* | 0.01 | [0,1] |
| $P_5$: reg_lambda | Booster | Regularization coefficient used to adjust *L2* | 0.1 | [0,2] |
| $P_6$: min_child_weight | Booster | Minimum leaf node weight | 1 | [1,10] |
| $P_7$: subsample | Booster | The sampling scale used for the training set | 1 | [0,1] |
| $P_8$: colsample_bytree | Booster | The random sampling ratio of features used to construct each tree | 1 | [0,1] |
| $P_9$: n_estimators | Learning Task | Controlling the number of trees | 70 | [30,200] |

In this study, the objective function of RS was set to the mean squared error (MSE) as the calculation error; the maximum number of iterations was set to 50; and the maximum value of MSE was set to 3.00. In the general parameters of XGBoost, the parameter "booster" was set to the "gbtree"; the parameter "n_thread" was set to not exceed the maximum possible number of threads of the processor; and other parameters were set to the default values. After the above optimized parameters were given the initialized default values, the processed data set was input into the optimization function of the RS, and the partial results of the optimization are shown in Table 3.

**Table 3.** Partial results of optimization of XGBoost.

| Iteration Number | $P_1$ | $P_2$ | $P_3$ | $P_4$ | $P_5$ | $P_6$ | $P_7$ | $P_8$ | $P_9$ | *MSE* |
|---|---|---|---|---|---|---|---|---|---|---|
| 1 | 5 | 0.2 | 0.01 | 0.01 | 0.1 | 1 | 1 | 1 | 70 | 2.54 |
| 2 | 12 | 0.009 | 0.526 | 0.635 | 0.137 | 6.23 | 0.972 | 0.892 | 102 | 2.23 |
| 3 | 9 | 0.417 | 0.391 | 0.172 | 0.935 | 5.33 | 0.391 | 0.913 | 61 | 1.78 |
| 4 | 16 | 0.103 | 0.153 | 0.229 | 0.182 | 6.19 | 0.672 | 0.992 | 94 | 1.52 |
| 5 | 8 | 0.319 | 0.203 | 0.381 | 0.286 | 4.27 | 0.259 | 0.715 | 105 | 0.936 |
| 10 | 11 | 0.821 | 0.337 | 0.156 | 0.107 | 8.31 | 0.113 | 0.836 | 82 | 0.783 |
| 16 | 7 | 0.113 | 0.082 | 0.096 | 0.132 | 4.92 | 0.463 | 0.651 | 103 | 0.497 |
| 22 | 10 | 0.016 | 0.128 | 0.193 | 0.092 | 1.37 | 0.991 | 0.192 | 75 | 0.585 |
| 50 | 3 | 0.432 | 0.11 | 0.308 | 0.087 | 1.86 | 0.723 | 0.274 | 91 | 0.647 |

By calculating the MSE, a more suitable parameter combination was found when the iteration number was 16th. However, the effect of the parameter combination gradually deteriorated with the number of iterations continuing to increase. Therefore, the parameters of the XGBoost were selected as the parameter combination in the 16th iteration to forecast APIIU of AQI and $PM_{2.5}$ in this research.

## 3. Analytical Results of APIIU

Before using SC analysis, based on the influence path of industrialization and urbanization on AQI and $PM_{2.5}$, this research constructed the structural equation modeling (SEM) using AMOS to evaluate the reasonableness of the model and preliminarily screen the relevant indicators from the indicators system [71,72]. Additionally, the least square

estimation method was selected to perform the parameter estimation of SEM. In this research, the comparative fit index (CFI), Akaike information criterion (AIC) and root mean square error of approximation (RMSEA) were selected as evaluation indicators. The model had better performance when CFI was higher than 0.8, RMSEA was lower than 0.1, and AIC was relatively small. Based on the structural equation and evaluation results, this research finally deleted the indicators with a moderating effect, including the square of coal consumption per land area, square of proportion of population and square of density of population. Hence, the above three indicators were no longer considered when using SC analysis. The SC correlation analysis results are shown in Figures 4 and 5.

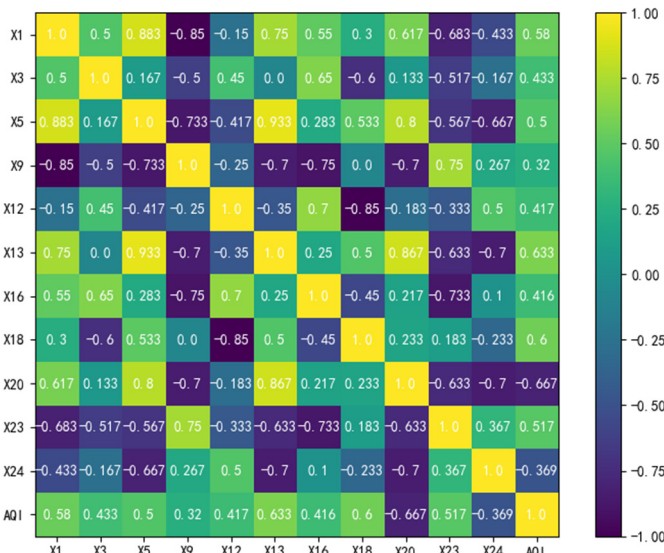

**Figure 4.** The SC correlation analysis results of APIIU of AQI.

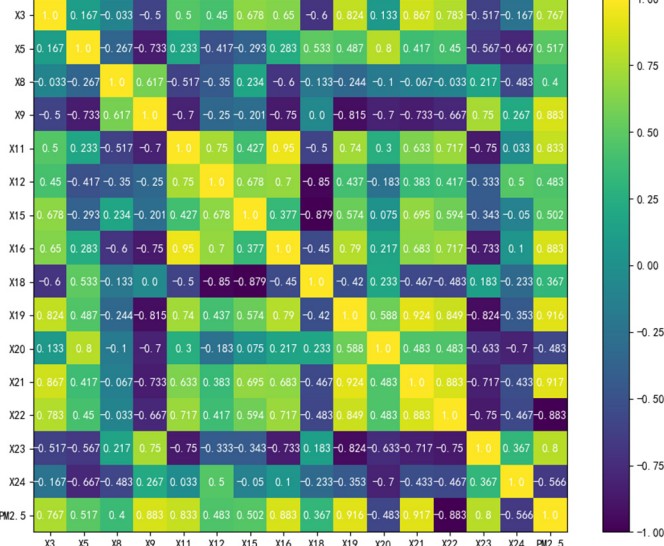

**Figure 5.** The SC correlation analysis results of APIIU of PM$_{2.5}$.

Before using SC to analyze the data, this research used the maximum and minimum standardized method in data processing, and the value of each indicator was controlled between 0 and 1. In order to obtain the weights of relevant indicators affecting the APIIU of AQI (APIIU-AQI) and APIIU of PM$_{2.5}$ (APIIU-PM$_{2.5}$), taking AQI and PM$_{2.5}$ as target variables, respectively, the correlation coefficients are shown in Figures 4 and 5. The indicators were from the indicators system shown in Table 1. In this research, the correlation coefficients of AQI and PM$_{2.5}$ were greater than 0.3, respectively. The correlation coefficients

in the SC analysis results reflected the degree of importance of the role of each indicator in the APIIU-AQI and AQPIIU-PM$_{2.5}$. Hence, the correlation coefficient of each indicator could be used to express the indicator weight. Using the weighting equation calculated using the air pollution index according to principal indicators, the APIIU could be expressed as in Equation (3):

$$I_i = \sum S_j X_{ij} \tag{3}$$

where $I_i$ is the APIIU of the $i_{th}$ unit with different principal indicators, $X_{ij}$ is standardized values of the $j_{th}$ indicator of the $i_{th}$ unit, and $S_j$ is the SC value of the $j_{th}$ indicator.

The APIIU-AQI and APIIU-PM$_{2.5}$ in every city calculated from the SC results showed that there were significant temporal and regional differences among the three provinces from 2017 to 2019. The changing trends in the APIIU-AQI and APIIU-PM$_{2.5}$ are shown in Figures 6a–c and 7a–c over the past 3 years, respectively. Considering the development of industrialization and urbanization, the APIIU-AQI and APIIU-PM$_{2.5}$ showed a significant changing trend from 2017 to 2019.

Based on the changing trend of APIIU-AQI and APIIU-PM$_{2.5}$ of every city within three provinces over the past 3 years, the following observations can be made. Hubei Province was an area with higher APIIU-AQI among the three provinces, indicating that the development of industrialization and urbanization had a more serious impact on AQI. From 2017 to 2019, the APIIU-AQI in 2019 in every city increased by 20–30% compared to 2017. Hunan Province was an area with higher APIIU-PM$_{2.5}$ among the three provinces, indicating that the development of industrialization and urbanization had a more serious impact on PM$_{2.5}$. From 2017 to 2019, the APIIU-PM$_{2.5}$ of every city within the three provinces over the past 3 years were higher compared to APIIU-AQI. In the past three years, the APIIU of some cities in Hunan Province has declined; however, some cities in Henan Province and Hubei Province have shown a trend of initial decline and then rise. The data showed a slowly decreasing trend and indicated that Hunan Province was paying attention to the protection and governance of the environment when developing its economy. Additionally, the industry has transformed into a new energy and environmentally friendly direction; the population density has decreased; and the air pollutant emissions emitted by human activities have decreased in Hunan Province. With an increasing trend in Henan Province and Hubei Province, this phenomenon showed that the proportion of secondary industry in the two provinces still occupied a major position. In addition, the increasing population and the government's control with a slow rate of environmental governance made the industrial and human activities aggravate environmental pollution. Through the assessment of APIIU-AQI and APIIU-PM$_{2.5}$ of regional air pollution and the descriptive analysis, the changing results of AQI and PM$_{2.5}$ were generally consistent with the findings published by the China Environmental Statistical Yearbook in Henan Province, Hubei Province and Hunan Province, respectively.

Air pollution has caused a serious impact on the natural environment, human society and residents' health in recent years. Hence, it is vital to evaluate the influence of the development of regional industrialization and urbanization on AQI and PM$_{2.5}$, which could help the government plan reasonable policies to a certain degree. For Henan Province, with a large population and agricultural industry, importance should be attached to increasing the development of environmental protection enterprises while transforming to heavy industrialization. It is necessary to avoid excessive population concentration during urbanization. For Hubei Province, with a high GDP, the pace of development of traditional industries can be slowed down while developing industrialization. The transformation to new energy enterprises can be intensified, which can not only protect the environment but prevent more population from outflowing. For Hunan Province, with the promotion of cleaner production technology and increasing financial investment in environmental protection, the environmental quality has been greatly improved. However, it should avoid repeating the same predicament. Environmental governance should be processed simultaneously while developing the economy and urbanization.

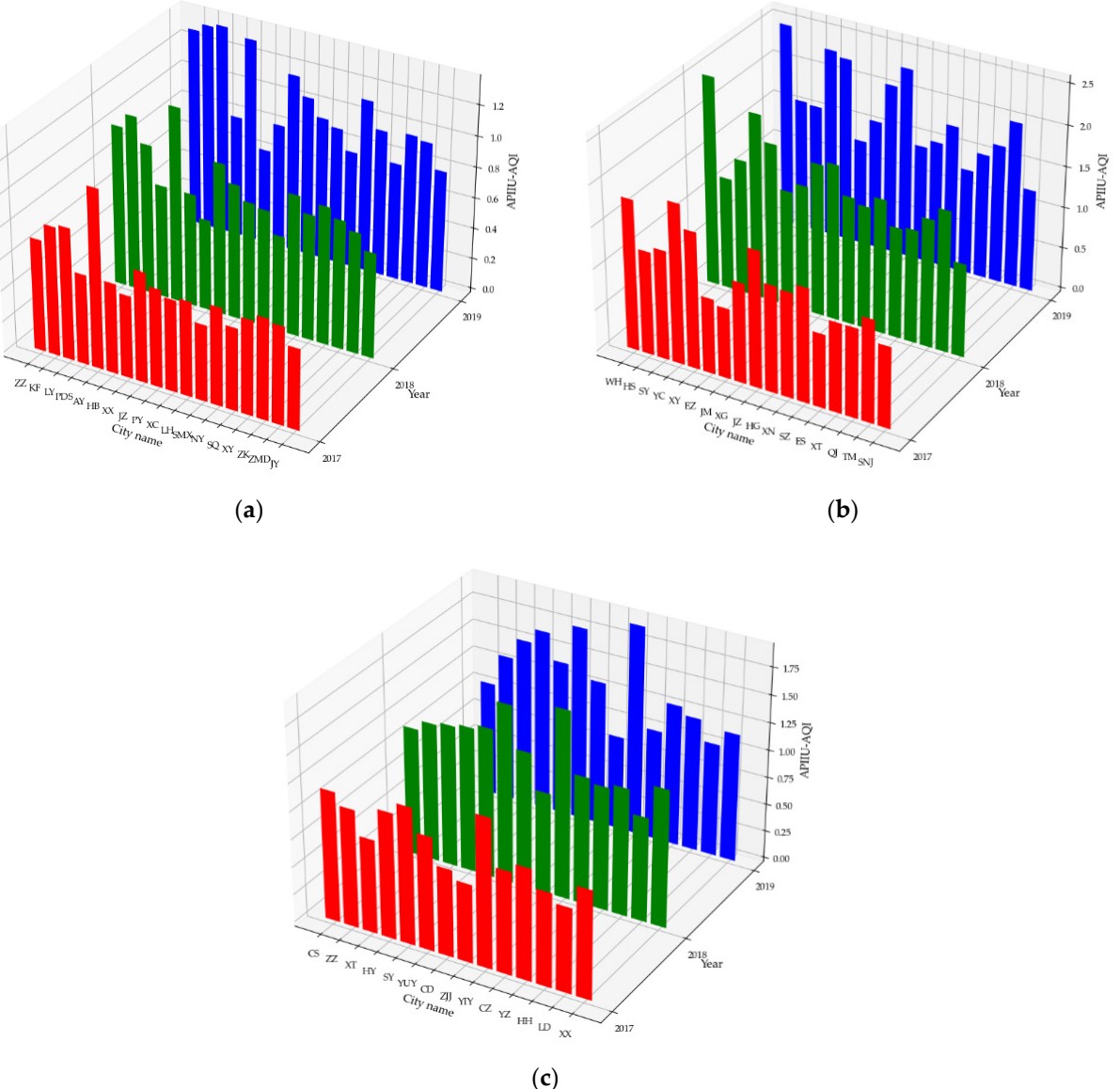

**Figure 6.** The changing trend of APIIU-AQI from 2017 to 2019: (**a**) Henan Province: Zhengzhou—ZZ, Kaifeng—KF, Luoyang—LY, Pingdingshan—PDS, Anyang—AY, Hebi—HB, Xinxiang—XX, Jiaozuo—JZ, Puyang—PY, Xuchang—XC, Luohe—LH, Sanmenxia—SMX, Nanyang—NY, Shangqiu—SQ, Xinyang—XY, Zhoukou—ZK, Zhumadian—ZMD, Jiyuan—JY; (**b**) Hubei Province: Wuhan—WH, Huangshi—HS, Shiyan—SY, Yichang—YC, Xiangyang—XY, Ezhou—EZ, Jingmen—JM, Xiaogan—XG, Jingzhou—JZ, Huanggan—HG, Xianning—XN, Suizhou—SZ, Enshi—ES, Xiantao—XT, Qianjiang—QJ, Tianmen—TM, Shennongjia— SNJ; (**c**) Hunan Province: Changsha—CS, Zhuzhou—ZZ, Xiangtan—XT, Hengyang—HY, Shaoyang—SY, Yueyang—YUY, Changde—CD, Zhangjiajie—ZJJ, Yiyang—YIY, Chenzhou—CZ, Yongzhou—YZ, Huaihua—HH, Loudi—LD, Xiangxi—XX.

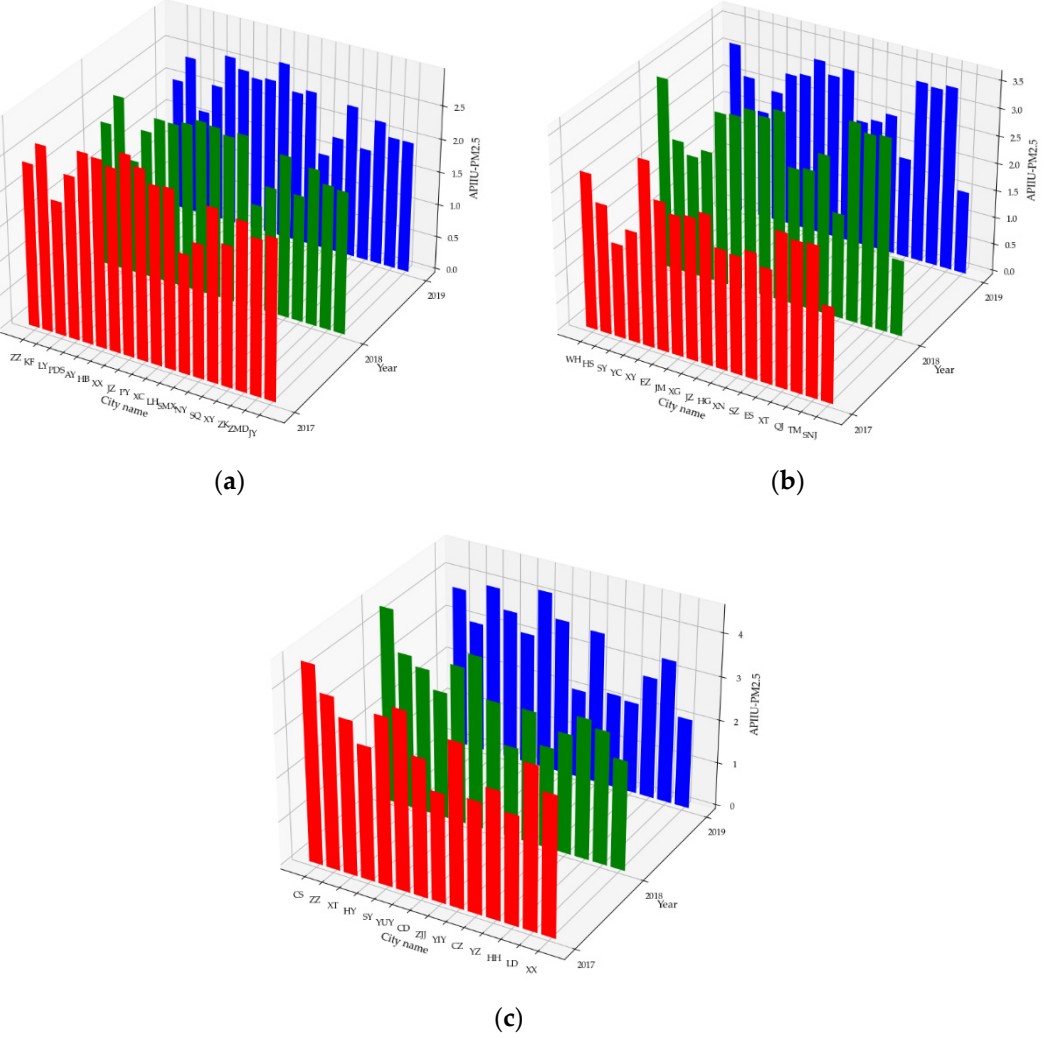

**Figure 7.** The changing trend of APIIU-PM$_{2.5}$ from 2017 to 2019: (**a**) Henan Province: Zhengzhou—ZZ, Kaifeng—KF, Luoyang—LY, Pingdingshan—PDS, Anyang—AY, Hebi—HB, Xinxiang—XX, Jiaozuo—JZ, Puyang—PY, Xuchang—XC, Luohe—LH, Sanmenxia—SMX, Nanyang—NY, Shangqiu—SQ, Xinyang—XY, Zhoukou—ZK, Zhumadian—ZMD, Jiyuan—JY; (**b**) Hubei Province: Wuhan—WH, Huangshi—HS, Shiyan—SY, Yichang—YC, Xiangyang—XY, Ezhou—EZ, Jingmen—JM, Xiaogan—XG, Jingzhou—JZ, Huanggan—HG, Xianning—XN, Suizhou—SZ, Enshi—ES, Xiantao—XT, Qianjiang—QJ, Tianmen—TM, Shennongjia— SNJ; (**c**) Hunan Province: Changsha—CS, Zhuzhou—ZZ, Xiangtan—XT, Hengyang—HY, Shaoyang—SY, Yueyang—YUY, Changde—CD, Zhangjiajie—ZJJ, Yiyang—YIY, Chenzhou—CZ, Yongzhou—YZ, Huaihua—HH, Loudi—LD, Xiangxi—XX.

## 4. Discussion

### 4.1. Evaluation Indicator

To reasonably evaluate the prediction model, the square of determination coefficient of real value ($R^2$), the root mean square error (RMSE) and the mean absolute percentage error (MAPE) were selected to test the prediction result. In statistical experiments, RMSE actually describes a degree of dispersion, which reflects the size of the average prediction error. MAPE can be used to evaluate different models on the same data source, which actually reflects the median relative error. The evaluation indicators can be shown as follows:

$$R^2 = 1 - \frac{\sum_{i=1}^{n}\left(y_i^* - y_i\right)^2}{\sum_{i=1}^{n}(y_i - \bar{y})^2} \tag{4}$$

$$RMSE = \frac{1}{n}\sqrt{\sum_{i=1}^{n}\left(y_i^* - y_i\right)^2} \tag{5}$$

$$MAPE = \sum_{i=1}^{n}\left|\frac{y_i^* - y_i}{y_i}\right| \times \frac{100}{n} \tag{6}$$

where $n$ represents the total number of test samples, $y_i^*$, $y_i$ and $\bar{y}$ represent the predicted value, the real value and the average real value in Equations (4)–(6), respectively.

### 4.2. Result Analysis Based on SC-RS-XGBoost

To verify the performance of the SC-RS-XGBoost model in predicting APIIU-AQI and APIIU-PM$_{2.5}$, this model was compared with the SC-XGBoost model, RS-XGBoost model and XGBoost model. For the SC-RS-XGBoost model and SC-XGBoost model, this research used SEM to screen the indicators and adopted the maximum and minimum method to standardize the indicators as the training data. Additionally, the original data were multiplied by the weights after SC analysis as the final training data. For the RS-XGBoost model and XGBoost model, this research used SEM to screen the indicators as the training data. Data were collected from 49 cities from 2017 to 2019, and the number of samples was 147. After the data were preprocessed, the feature data and target data were input into the different models, and the ratio of training data and testing data was set to 3:1. APIIU-AQI and APIIU-PM$_{2.5}$ were selected as the target variables, respectively. Industrialization, urbanization and meteorological indicators were selected as the characteristic variables. The predicted values of different models are shown in Figures A1 and A2 in Appendix A. Collecting the prediction results and calculating the evaluation indicators, the results are shown in Table 4.

**Table 4.** Evaluation indicators of APIIU of each model.

| Variable | SC-RS-XGBoost | | | SC-XGBoost | | |
|---|---|---|---|---|---|---|
| | $R^2$ | RMSE | MAPE | $R^2$ | RMSE | MAPE |
| APIIU-AQI | 0.945 | 0.103 | 4.25% | 0.886 | 0.149 | 6.08% |
| APIIU-PM$_{2.5}$ | 0.897 | 0.205 | 4.84% | 0.515 | 0.443 | 6.34% |
| **Variable** | **RS-XGBoost** | | | **XGBoost** | | |
| | $R^2$ | RMSE | MAPE | $R^2$ | RMSE | MAPE |
| APIIU-AQI | 0.856 | 0.167 | 6.36% | 0.753 | 0.218 | 13.7% |
| APIIU-PM$_{2.5}$ | 0.823 | 0.269 | 5.74% | 0.366 | 0.646 | 16.8% |
| **Variable** | **SC-RS-SVR** | | | **SC-SVR** | | |
| | $R^2$ | RMSE | MAPE | $R^2$ | RMSE | MAPE |
| APIIU-AQI | 0.503 | 0.285 | 7.13% | 0.377 | 0.514 | 8.83% |
| APIIU-PM$_{2.5}$ | 0.419 | 0.912 | 10.7% | 0.283 | 2.63 | 14.6% |
| **Variable** | **RS-SVR** | | | **SVR** | | |
| | $R^2$ | RMSE | MAPE | $R^2$ | RMSE | MAPE |
| APIIU-AQI | 0.412 | 0.393 | 13.5% | 0.461 | 0.427 | 17.2% |
| APIIU-PM$_{2.5}$ | 0.318 | 1.51 | 7.32% | 0.252 | 2.74 | 16.6% |

Using SC, reducing the input variables and reducing the target function convergence value effectively solved the correlation between the input variables and the defection of excessive input data. In this paper, RF was used to optimize the parameters of XGBoost to reduce local overfitting, showing better prediction performance. It can be seen from the results that SC-RS-XGBoost performed better than SC-XGBoost. As shown in Table 4, we found that the traditional XGBoost model had the largest prediction error. As seen from the comparison of $R^2$, *RSME* and MAPE of APIIU-AQI, the prediction accuracy of the SC-RS-XGBoost model improved by 6.24%, 30.9% and 30.1% upon the SC-XGBoost

model on the $R^2$, RMSE and MAPE, respectively. It improved by 8.90%, 38.3% and 33.1% upon the RS-XGBoost model on the $R^2$, RMSE and MAPE, respectively, and it improved by 20.3%, 52.8% and 68.9% upon the traditional XGBoost model on the $R^2$, RMSE and MAPE, respectively. For a comparison of $R^2$, $RSME$ and MAPE of APIIU-PM$_{2.5}$, the prediction accuracy of the SC-RS-XGBoost model improved by 42.6%, 53.7% and 23.7% upon the SC-XGBoost model on the $R^2$, RMSE and MAPE, respectively. It improved by 8.24%, 23.8% and 15.7% upon the RS-XGBoost model on the $R^2$, RMSE and MAPE, respectively, and it improved by 59.2%, 68.3% and 71.2% upon the traditional XGBoost model on the $R^2$, RMSE and MAPE, respectively. Based on these findings, the prediction accuracy was significantly improved, which verified the effectiveness and feasibility of the SC-RS-XGBoost model compared with the other three models. To further verify the performance and accuracy, the evaluation indicators with the SC-RS-SVR model, SC-SVR model, RS-SVR model and SVR model are shown in Table 4. The comparisons of $R^2$, RMSE and MAPE of the SC-RS-SVR model, SC-SVR model, RS-SVR model, SVR model and ELM model are shown in Figures S1–S3. It can be seen from the calculation results of evaluation indicators that other models did not perform as well as the SC-RS-XGBoost model.

Additionally, through the calculation results of evaluation indicators, we found that the prediction accuracy of APIIU-AQI was higher than APIIU-PM$_{2.5}$. In terms of RMSE and MAPE, the difference between the APIIU-AQI and APIIU-PM$_{2.5}$ was also basically inconspicuous. However, the values of $R^2$ were all higher than 0.89, showing that the SC-RS-XGBoost model had a good fitting performance on APIIU-AQI and APIIU-PM$_{2.5}$. By verifying the effectiveness and feasibility of the proposed model, the SC-RS-XGBoost model accurately predicted the APIIU-AQI and APIIU-PM$_{2.5}$ and provided helpful insights for the government's environmental protection and air pollution governance. This could make regional governments pay more attention to air pollution when developing industrialization and urbanization.

## 5. Conclusions

Most previous research works have analyzed AQI and PM$_{2.5}$ spatiotemporal distribution using statistical methods and regression analyses to evaluate the air quality. The goal of the current research was to analyze and determine the influence of the development of industrialization and urbanization on AQI and PM$_{2.5}$. In this period of artificial intelligence, a SC-RS-XGBoost model for air pollution assessment could effectively improve the assessment accuracy and provide a new reference source for future air pollution management. Based on previous relevant research works, this research constructed a regional air pollution assessment indicators system to evaluate APIIU-AQI and AQIIU-PM$_{2.5}$ using data from Henan Province, Hubei Province and Hunan Province. The indicators system of this research adopted three aspects, including industrialization indicators, urbanization indicators and meteorological indicators. The following conclusions were drawn from our findings:

(1) The principal indicators were screened using SC, and the weights of each indicator were determined according to the correlation coefficients. From the correlation analysis, there were differences in the impact of different indicators on AQI and PM$_{2.5}$. The indicators, including regional GDP, proportion of secondary industry, coal consumption, density of exhaust emissions, total city population, proportion of population, density of population and several relevant meteorological indicators, had a high influence on AQI. However, the proportion of secondary industry, coal consumption, exhaust emissions, density of exhaust emissions, population of city jurisdiction, total city population, administrative land areas and other related indicators had a high influence on PM$_{2.5}$.

(2) Using the weights equation to calculate the APIIU-AQI and APIIU-PM$_{2.5}$ of each region from 2017 to 2019, we identified that 17 cities from Hubei Province, which has the largest regional GDP and the largest proportion of the population, had the largest APIIU-AQI in the past three years. Moreover, APIIU-AQI increased sharply compared with 2017 in the three provinces. Fourteen cities from Hunan Province, which has the largest

proportion of secondary industry and the largest administrative land area, had the largest APIIU-PM$_{2.5}$ in the past three years. APIIU-PM$_{2.5}$ has gradually increased compared with 2017 in the three provinces.

(3) This research verified that the SC-RS-XGBoost could be used as a method of air pollution assessment. Analyzing the influence of industrialization and urbanization on AQI and PM$_{2.5}$ is of great significance for future regional sustainable development. It is necessary for carrying out more advanced research and constructing a more accurate indicators system and prediction models for the various different regions to deliver useful information for the government officials and policymakers.

The present research contributed to the limited knowledge in the regions regarding the influence of industrialization and urbanization on spatiotemporal variations of AQI and PM$_{2.5}$ in three Chinese provinces, and it is important to conduct additional research in other regions on this topic. AQI and the concentration of PM$_{2.5}$ in different regions are affected by wind speed and direction at different altitudes to some extent. It is worth noting that this research was focused only on static and mild wind conditions, and the annual average wind speed was only observed at 70 m–80 m altitude. The result was limited, since the data selected for discussion were monitored under specific climate conditions. Hence, long-term continuous monitoring under various weather conditions is needed in future research. Due to differences in industrial and human activity emissions in different seasons, in order to explore the seasonal changes, it is necessary to consider the influence of industrialization and urbanization indicators on AQI and PM$_{2.5}$ in different seasons in future research.

**Supplementary Materials:** The following supporting information can be downloaded at: https://www.mdpi.com/article/10.3390/atmos13091377/s1, Table S1: The statistical analysis of PM$_{2.5}$ in 2017 in three provinces; Table S2: The statistical analysis of PM$_{2.5}$ in 2018 in three provinces; Table S3: The statistical analysis of PM$_{2.5}$ in 2019 in three provinces; Figure S1: Comparison of $R^2$ of SC-RS-SVR model, SC-SVR model, RS-SVR model, SVR model and ELM model; Figure S2: Comparison of RMSE of SC-RS-SVR model, SC-SVR model, RS-SVR model, SVR model and ELM model; Figure S3: Comparison of MAPE of SC-RS-SVR model, SC-SVR model, RS-SVR model, SVR model and ELM model.

**Author Contributions:** Conceptualization, H.C.; methodology, H.C.; validation, G.D. and Y.L.; formal analysis, H.C.; investigation, H.C.; resources, H.C.; data curation, H.C.; Writing—Original draft preparation, H.C.; Writing—Review and editing, H.C.; supervision, G.D.; project administration, H.C. All authors have read and agreed to the published version of the manuscript.

**Funding:** This research was greatly supported by the National Social Science Fund Key Project (15AGL013); Henan Provincial Department of Science and Technology Risk Management Innovation and Public Policy Soft Science Research Base, Henan Social Science Planning Project (2019BJJ030). Research on the construction of disaster prevention and reduction support system in large and medium-sized cities in Henan Province (22240041001); Henan Provincial Colleges and Universities Philosophy and Social Science Basic Research Major Project "Evaluation Research on Comprehensive Disaster Resilience Capacity of Chinese Communities" (2021JCZD04).

**Institutional Review Board Statement:** Not applicable.

**Informed Consent Statement:** Not applicable.

**Data Availability Statement:** Not applicable.

**Conflicts of Interest:** The authors declare no conflict of interest.

## Appendix A

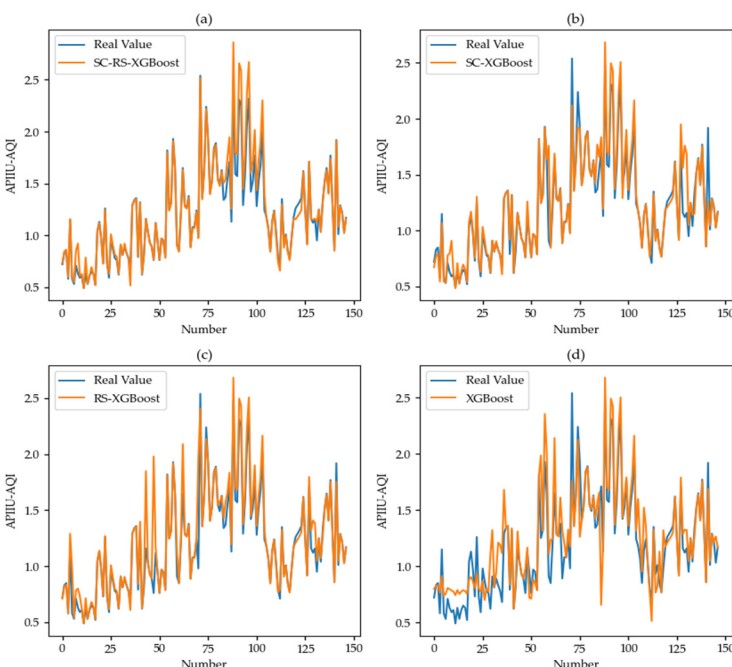

**Figure A1.** Comparison of real values and predicted values of APIIU-AQI from (**a**) SC-RS-GBoost model, (**b**) SC-XGBoost model, (**c**) RS-XGBoost model and (**d**) XGBoost model.

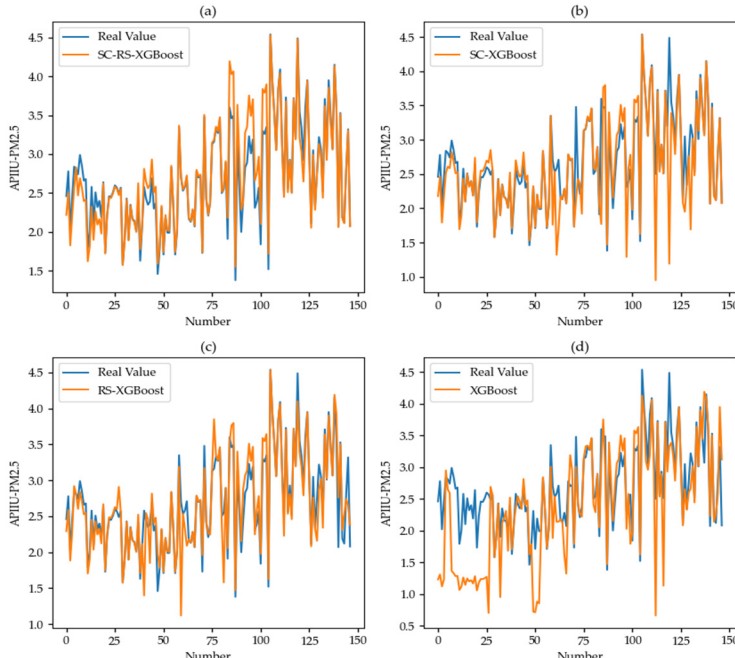

**Figure A2.** Comparison of real values and predicted values of APIIU-PM$_{2.5}$ from (**a**) SC-RS-XGBoost model, (**b**) SC-XGBoost model, (**c**) RS-XGBoost model and (**d**) XGBoost model.

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
