# Peer review of "Monitoring the Influence of Industrialization and Urbanization on Spatiotemporal Variations of AQI and PM2.5 in Three Provinces, China"

_atmosphere, doi:10.3390/atmos13091377_

Round 1

Reviewer 1 Report

Recommendation

The article analyze and determine the influence of the development of Industrialization, urbanization and metrology on AQI and PM2.5 using machine learning methods in 3 Chinese provinces during 2017-2019. The manuscript discuss the important factors influence on air quality and well structure but need author to improve the quality and thus required major revision.

1: The title should be revised as does not look like a professional style.

 For example; Monitoring the Influence of Industrialization and Urbanization on spatiotemporal variations of AQI and PM2.5 in Three Provinces, China

2: Need to restructure abstract and add key results with values. Model accuracy, R² and average AQI and PM2.5 values etc

3: Found basic level mistakes;

• Line 13, 120-122, 245-247, and 483-484: indicator are 3 or 4 times used, need to limit to one time

• Why using abbreviation if it is not in use along further text sections: World Health Organization (WHO) and European Union (EU)

• Use full form first then use abbreviation further along text for gross domestic product (GDP)

• Definition of PM2.5 and PM10 are not correctly provided

• In some cases PM2.5, 2.5 need to be subscript from Figure 7 caption, table 4 and last pages

• Line 169 and 170 both cite the same website. Recommended to remove first citation of website link, 2nd one is enough.

• Figure 1: 3 provinces map required same size as China map and include longitude and latitude information

• It’s recommended to merge 3 provinces maps set in figure 2 instead of separate maps

• Line 177: please indicate other tools used in this study with version

4: Although work is good for basic understanding. There should be the categorization of data on a seasonal based for deep understanding. I shall suggest to draw more interesting map based on seasonal variation.

5: As data shows Industrialization and Urbanization result in air pollution. What is your recommendation for policymakers to adopt for sustainable development?

6: Figure A2 is for APIIU-PM2.5 or AQI, need to correct figure

7: In supplementary tables, sum of values not required. Instead percentile and number of values will be better to be added.

Author Response

Response of

“Monitoring the Influence of Industrialization and Urbanization on Spatiotemporal Variations of AQI and PM2.5 in Three Provinces, China”

Dear Reviewer,

Thank you for reviewing our paper. The new title of this paper is "Monitoring the Influence of Industrialization and Urbanization on Spatiotemporal Variations of AQI and PM2.5 in Three Provinces, China" authored by Hu Chen, Guoqu Deng and Yiwen Liu. We are grateful for the feedback of you, which has led to an improvement of our paper. We list our responses (in blue regular font). All the modifications are highlighted in red regular font in the new version of our manuscript:

1: We have revised the original title “Research on the Influence of Industrialization and Urbanization on Air Quality Index and PM2.5 in Three Provinces, China”.

2: We have revised basic level mistakes according to your report. For example, we have limited the indicator to be one time used, deleted the abbreviation of World Health Organization (WHO) and European Union (EU), added the full form gross domestic product (GDP), 2.5 in Figure 7 caption, Table 4 and last pages have been revised, and indicated other tools used in this study with version. At last, Figure A2 for APIIU-PM2.5 has been revised.

3: We have added the evaluation indicator of R2 revised in the whole manuscript, and provided the definition of PM2.5 and PM10 correctly in the appropriate place (In the section of 2.1.3. Indicators System).

4: We have revised the regional map (Figure 1) and the annual average AQI changing trend (Figure 2).

5: According to the changing trends of AQIIU-AQI and AQIIU-PM2.5, we have provided recommendations for policymakers in every province to adopt for sustainable development. In the section of 3. analytical Results of APIIU, we have added the recommendations.

6: The sum of values of PM2.5 in the supplementary material were deleted.

We reconsidered the main objective of the paper. We carefully remodified many sentences and paragraphs. Analysis numbers in manuscript and tables have been remodified. Spearman coefficient (SC) correlation analysis, a random search (RS) algorithm and an excellent algorithm extreme gradient boosting (XGBoost) to evaluate the air pollution influence of industrialization and urbanization (APIIU). Industrialization, urbanization and meteorological indicators were used to measure the influence degree of APIIU on AQI and particulate matter 2.5 (PM2.5), respectively. Because the time span of our research was annual data, the data we collect was calculated as annual data and we didn't consider using quarterly data. The target variables and characteristic variables of this research were also predicted based on the annual data of each province. In supplementary tables, sum of values have been deleted. These data were for the purpose of supplementing the annual data of PM2.5, and the data of a total of 49 cities in three provinces were selected, 18 cities in Henan Province, 17 cities in Hubei Province and 14 cities in Hunan Province.

Special thanks for your comments. We have tried our best to improve the manuscript according to the comments. We appreciate for your warm work earnestly and hope the revision would meet with approval. Once again, thank you very much for your comments and suggestions.

Best wishes.

Author Response

Response of

“Monitoring the Influence of Industrialization and Urbanization on Spatiotemporal Variations of AQI and PM2.5 in Three Provinces, China”

Dear Reviewer,

Thank you for reviewing our paper. The new title of this paper is "Monitoring the Influence of Industrialization and Urbanization on Spatiotemporal Variations of AQI and PM2.5 in Three Provinces, China" authored by Hu Chen, Guoqu Deng and Yiwen Liu. We are grateful for the feedback of you, which has led to an improvement of our paper. We list our responses (in blue regular font). All the modifications are highlighted in red regular font in the new version of our manuscript:

1: We have revised introduction section.

2: We have using boxplots to describe the PM2.5 in Figure 3.

3: We have revised the section of 2.2.3. We have added the basic process of SC-RS-XGBoost algorithm. In addition, we put the basic introduction of XGBoost in the supplementary material.

4: We have added give explanation about the reason that SC-RS-XGBoost performed better than SC-XGBoost. Also, we have added give explanation about the reason that APIIU-AQI were lower than those from APIIU-PM2.5.

We have added a brief process description of the overall study and the origin of the method citation in introduction section. This study chooses to use a heat map to display the AQI, so that readers can more clearly see the change trend of air quality in terms of time and space in three years. Therefore, Figure 3 was modified and Figure 2 was’t modified. We adopt the basic processing flow of SC-RS-XGBoost as the mathematical expression of the algorithm.

Special thanks for your comments. We appreciate for your warm work earnestly and hope the revision would meet with approval. Once again, thank you very much for your comments and suggestions.

Best wishes.

Reviewer 3 Report

Good work, described in detail both in the methods used and the results obtained.

Author Response

(The authors gave the same response as above.)

Round 2

Reviewer 1 Report

The quality of paper is much improved and appreciated. Only a couple of minor things to consider by authors before final publication. 

1) Need to add limitations of current study i.e., seasonal variations etc.

2) Line 179: Have you used ArcGIS, if yes include it with version

3) Reconsider heading 4.2 text. And its suggested to show results of figure 8-10 in table 4. Figure can be moved in supplementary / appendix section with addition of XGBoost data (all models)

4) In some cases PM2.5 "2.5" need to be subscript

Author Response

Response of

“Monitoring the Influence of Industrialization and Urbanization on Spatiotemporal Variations of AQI and PM2.5 in Three Provinces, China”

Dear Reviewer,

Thank you for reviewing our paper. The title of this paper is "Monitoring the Influence of Industrialization and Urbanization on Spatiotemporal Variations of AQI and PM2.5 in Three Provinces, China" authored by Hu Chen, Guoqu Deng and Yiwen Liu. We are grateful for the feedback of you, which has led to an improvement of our paper. We list our responses (in blue regular font). All the modifications are highlighted in red regular font in the new version of our manuscript:

1: We have added the version of ArcGIS 10.8 in the line 179-180.

2: We have revised "2.5", which needed to be subscript in some cases.

3: We have revised the heading 4.2 text. Figure 8-10 were moved into the supplementary material.

4: We have added the limitations of seasonal variations in this research.

Special thanks for your comments. We appreciate for your warm work earnestly and hope the revision would meet with approval. Once again, thank you very much for your comments and suggestions.

Best wishes.
